# Ectopic Expression of Neurod1 Is Sufficient for Functional Recovery following a Sensory–Motor Cortical Stroke

**DOI:** 10.3390/biomedicines12030663

**Published:** 2024-03-15

**Authors:** Jessica M. Livingston, Tina T. Lee, Tom Enbar, Emerson Daniele, Clara M. Phillips, Alexandra Krassikova, K. W. Annie Bang, Ines Kortebi, Brennan W. Donville, Omadyor S. Ibragimov, Nadia Sachewsky, Daniela Lozano Casasbuenas, Arman Olfat, Cindi M. Morshead

**Affiliations:** 1Department of Surgery, Division of Anatomy, 1 King’s College Circle, University of Toronto, Toronto, ON M5S 1A8, Canada; jmlivingston@hollandcollege.com (J.M.L.); emerson.daniele@mail.utoronto.ca (E.D.); ines.kortebi@mail.utoronto.ca (I.K.);; 2Donnelly Centre for Cellular and Biomolecular Research, University of Toronto, 160 College Street, Toronto, ON M5S 3E1, Canada; 3Institute of Medical Science, 1 King’s College Circle, University of Toronto, Toronto, ON M5S1A8, Canada; 4Lunenfeld-Tanenbaum Research Institute, 600 University Ave., Toronto, ON M5G 1X7, Canada; bang@lunenfeld.ca; 5Institute of Biomedical Engineering, 1 King’s College Circle, University of Toronto, Toronto, ON M5S 1A8, Canada

**Keywords:** direct lineage conversion, *Neurod1*, brain repair, stroke, functional recovery, ectopic transcription factor expression, adeno-associated virus, astrocyte-to-neuron conversion, neuroregeneration, gait analysis

## Abstract

Stroke is the leading cause of adult disability worldwide. The majority of stroke survivors are left with devastating functional impairments for which few treatment options exist. Recently, a number of studies have used ectopic expression of transcription factors that direct neuronal cell fate with the intention of converting astrocytes to neurons in various models of brain injury and disease. While there have been reports that question whether astrocyte-to-neuron conversion occurs in vivo, here, we have asked if ectopic expression of the transcription factor *Neurod1* is sufficient to promote improved functional outcomes when delivered in the subacute phase following endothelin-1-induced sensory–motor cortex stroke. We used an adeno-associated virus to deliver *Neurod1* from the short GFAP promoter and demonstrated improved functional outcomes as early as 28 days post-stroke and persisting to at least 63 days post-stroke. Using *Cre*-based cell fate tracking, we showed that functional recovery correlated with the expression of neuronal markers in transduced cells by 28 days post-stroke. By 63 days post-stroke, the reporter-expressing cells comprised ~20% of all the neurons in the perilesional cortex and expressed markers of cortical neuron subtypes. Overall, our findings indicate that ectopic expression of *Neurod1* in the stroke-injured brain is sufficient to enhance neural repair.

## 1. Introduction

Stroke is a leading cause of death and disability worldwide, and many survivors experience long-lasting impairments [1]. Approximately 850,000 individuals in North America suffer from strokes annually [2]. Survivors often experience motor impairments, particularly affecting sensory-motor function in the upper limbs [2]. This can lead to chronic disability that significantly diminishes quality of life and increases the risk of post-stroke depression [3]. With an aging population and increasing projected incidence of stroke [4], there is an urgent need for alternative options to improve post-stroke impairments.

Numerous approaches to treat stroke have been explored, including activation of endogenous neural stem cells [5,6,7] and cell transplantation [8,9]. Since the discovery that ectopic transcription factor (TF) expression can drive cell fate [10], attention has been focused on developing interventions to replace lost cells in models of injury or disease. Many publications have since identified effective methods to convert somatic cells to other cell types [11,12], including neuronal cells, both in vitro and in vivo [13,14,15,16,17,18,19,20,21,22]. In the context of the central nervous system (CNS), a number of groups have examined the ectopic expression of TFs in the CNS and demonstrated that proneural TFs such as Ngn2 [23] and Ascl1 [20,24] and the neuronal differentiation TF Neurod1 [25,26] can convert glial cells to neurons [27]. Further, studies have shown that induced neurons from the ectopic expression of TFs have regionally appropriate [14,28,29] and functional neuronal subtype identities [15,30,31,32,33]. Some recent studies in the field have brought into question the efficacy of in vivo conversion [34,35], and this has served to highlight the importance of addressing potential caveats to further advance the field [19,36,37]. Wang et al. [34] reported off-target neuronal expression of adeno-associated virus (AAV)-based delivery strategies driving TF expression from the short human GFAP promoter (shGFAP). While the mechanism is not entirely understood, the implication is that transduced endogenous neurons could mistakenly be interpreted as astrocyte-to-neuron reprogramming [34,35]. Nevertheless, the report by Wang et al. [34] does not negate the capacity of glia to be converted to neurons [27,38,39].

Of particular interest in developing therapeutics to promote neural repair, a number of groups have validated the feasibility of ectopic TF expression in preclinical models of injury and disease, including Alzheimer’s disease [40], Parkinson’s disease [41], and stroke [26,42]. These studies have focused on cellular outcomes, and a considerable gap exists in understanding the resulting functional outcomes of the AAV-mediated ectopic expression of neural TFs. Acknowledging that endogenous neurons may be targeted to overexpress TFs delivered by AAVs underscores the possibility that ectopic expression of TFs in neurons may underlie any improved outcomes, making this approach promising to treat neurological disease. Hence, it is important to study the functional outcomes of forced TF expression-based therapy. To this end, in a study of Parkinson’s disease, the ectopic expression of a cocktail of TFs that promote dopaminergic cell fate in astrocytes improved some aspects of motor function [41]. Further, improved functional outcomes were recently reported using the TF *Neurod1* in a rodent ischemia model [42]. While these studies highlight the promising potential of gene therapy for neurorepair and functional recovery there is caution when advancing the translation of preclinical studies. For example, translating preclinical stroke research has not proven successful, and a number of recommendations and guidelines have been put forth by the Stroke Therapy Academic Industry Roundtable (STAIR) to overcome this challenge, including the rigorous testing and replication of potential therapeutics in more than one preclinical model [43].

Herein, we aimed to understand the longer-term functional outcomes of ectopic TF expression. We used AAV-mediated delivery of *Neurod1* to GFAP-expressing cells in a well-established preclinical model of sensory-motor ischemic stroke. The endothelin-1 model [44,45] has the advantage of generating highly reproducible lesions and controlled and persistent deficits in upper limb function, making it useful for studying post-stroke recovery strategies. Different from the published stroke studies of Chen and colleagues [42], we chose a different AAV serotype to deliver *Neurod1* based on reports suggesting enhanced astrocyte targeting with AAV5 [46]. Further, the previous stroke study used the ubiquitous CAG promotor to drive TF expression. Herein, *Neurod1* expression was under the control of the short GFAP (gfaAVC(1)D) promotor to further target astrocyte-driven expression. We performed our experiments in a conditional reporter mouse that enabled us to track transduced cells and performed functional assays up to 9 weeks post-stroke. We found that ectopic expression of *Neurod1* delivered in a sub-acute phase post-cortical stroke led to persistent improved motor outcomes. Virally transduced cells expressed cortical layer-appropriate neuronal markers. These findings provide further support that ectopic expression of *Neurod1* is a viable approach to improve stroke recovery.

## 2. Materials and Methods

### 2.1. Mice 

Adult (8–12 week old) male and female R26R-YFP (Jackson Labs, Bar Harbor, ME, USA: B6.129X1-*Gt(ROSA)26Sortm1(EYFP)Cos*/J) and R26R-tdTomato (Jackson Labs: *Gt(ROSA)26Sor ^tm9(CAG-tdTomato)Hze^*) mice were used in this study [47]. Mice were group-housed in a barrier facility with a 12 h light/12 h dark cycle with ad libitum access to food and water. Experiments were conducted according to protocols approved by the Institutional Animal Care Committee and performed in accordance with guidelines published by the Canadian Council for Animal Care.

### 2.2. Endothelin-1 Stroke

The vasoconstricting peptide endothelin-1 (ET-1; Calbiochem, San Diego, CA, USA) was used to induce focal stroke to the sensory–motor cortex as previously described [9,48]. Mice were anesthetized using isofluorane and mounted onto a stereotaxic apparatus. The scalp was incised, and a small hole was drilled into the skull at the injection location. A total of 1 μL of 400 picomolar ET-1 dissolved in sterile PBS was injected into the sensory–motor cortex at +0.6 AP, +2.20 ML, and −1.0 DV from bregma. A 26-gauge Hamilton syringe with a 45-degree beveled tip was used to inject ET-1 at a rate of 0.1 μL/min. The needle was left in place for 10 min after completion of the ET-1 injection to prevent backflow, then slowly withdrawn. Body temperature was maintained at 37 °C using a heating pad, and animals recovered under a heat lamp. Ketoprofen (5.0 mg/kg; s.c.) was administered for post-surgery analgesia.

### 2.3. Adeno-Associated Virus (AAV) Injections

AAV5-GFAP(0.7)^promoter(gfaABC(1)D)^::*Neurod1*-T2A-*Cre*-WPRE (AAV5::*Neurod1*; 5.0 × 10^12^ GC/mL) and AAV5-GFAP(0.7)^promoter(gfaABC(1)D)^-*Cre*-WPRE (AAV5::Cre; 1.1 × 10^13^ GC/mL) were generated and packaged by Vector BioLabs (Malvern, PA, USA). 

Mice were anesthetized with isofluorane and mounted onto a stereotaxic apparatus. A total of 1 μL of AAV5::*Neurod1* or AAV5::*Cre* was injected into the ipsilesional cortex 7 days post-stroke or -sham injury, at a rate of 0.1 μL/min at each the following coordinates: [+0.6 AP, +2.2 ML, −1.0 DV], [+1.6 AP, +2.2 ML −1.0 DV], and [−0.4 AP, +2.2 ML, −1.0 DV] mm from bregma, representing regions encompassing the anterior-to-posterior extent of the stroke-lesioned brain. The needle was left in place for 10 min after each AAV injection to prevent backflow and then slowly withdrawn. Body temperature was maintained at 37 °C using a heating pad, and animals recovered under a heat lamp. Ketoprofen (5.0 mg/kg; s.c.) was administered for post-surgery analgesia. 

### 2.4. Tissue Processing, Immunohistochemistry, Imaging, and Quantification

Mice were anesthetized with tribromoethanol (Avertin; 250 mg/kg; i.p.) and perfused transcardially with saline, followed by 4% paraformaldehyde (PFA). Brains were removed, post-fixed for 4 h with 4% PFA, and then placed in 30% sucrose to cryopreserve. Brains were frozen, sectioned (20 µm) using a cryostat, mounted onto Superfrost Plus slides, and stored at −20 °C.

For immunohistochemistry, slides were sectioned, blocked with 10% normal goat serum (NGS) and 0.3% Triton x-100 in PBS, and labeled with primary antibodies (for astrocytes, rabbit-anti-GFAP was used, 1:3000, DAKO, Santa Clara, CA, USA, 2016-04; or chicken-anti-GFAP, 1:1000, Aves, GFP-1020. For neurons, rabbit-anti-NeuN was used, 1:100, Millipore, Oakville, ON, Canada, ABN78. For layer-specific markers, the upper-layer marker rabbit-anti-CUX1 was used, 1:100, Abcam, Baltimore, MA, USA, AB54583; and the lower-layer marker rat-anti-CTIP2, 1:300, Abcam, AB18465. To detect evidence of activity, the early immediate gene protein mouse-anti-cFOS was used, 1:200, Santa Cruz, Dallas, TX, USA, Sc-166940. To determine the cell phenotype as oligodendrocyte, microglia, or migratory neuroblasts, Olig 2, 1:500, Millipore, Iba1, 1:500, Wako, Richmond, VA, USA were used) in PBS overnight at 4 °C, followed by incubation with secondary antibodies (1:400, from Life Technologies (Carlsbad, CA, USA): goat-anti-rabbit 568; goat-anti-rabbit 647; goat-anti-chicken 488; goat-anti-mouse 488) and DAPI (1:10,000; Invitrogen, Waltham, MA, USA) in PBS for 1 h at room temperature. Slides were then cover-slipped using Dako fluorescent mounting media (ThermoFisher, Waltham, MA, USA). 

Imaging was performed using Zen 2011 (v.1.0) software and a Zeiss confocal microscope (LSM880, v.14, Zeiss, Oberkochen, Germany). Linear contrast/brightness adjustments were performed using Zen 2011 software or Adobe Photoshop (v.20). Colocalization analysis was performed by *z*-stack analysis to confirm overlapping expression in confocal images in three images from three different coronal sections (located at +1.6, +0.6, and −1.0 AP from bregma) per animal viewed at 20× magnification. Identical linear adjustments of contrast and brightness were made to micrographs using the respective microscope software or Photoshop using the levels function, as follows: CUX1 (linear brightness adjustment, all channels), TBR1 (linear brightness adjustment, red channel), CTIP2 (linear brightness adjustment, red channel), and c-FOS (linear brightness adjustment, green channel).

### 2.5. Lesion Volume Analysis

Lesion volume was calculated from NeuN-stained sections and defined as areas devoid of NeuN+ stain. Image J (v.2, National Institute of Health, New York, NY, USA) was used to measure this area in 5 × 20 µm thick coronal sections (160 µm apart) spanning the anterior–posterior extent of the injury site. The software was calibrated using a scale bar, and the area measurement was obtained using the freehand selection tool. Area was then multiplied by the distance between sections to estimate the total infarct volume.

### 2.6. Astrocyte Sorting and ImageStream Analysis

Mice were anesthetized with isofluorane and cervically dislocated. The brain and meninges were removed, and coronal slices were obtained using a scalpel blade. The cortex was collected from slices that included the lesioned cortex, taking care to avoid the corpus callosum. The tissue was enzymatically and mechanically dissociated into a single-cell suspension using the magnetically activated cell-sorting (MACS) Neural Tissue Dissociation Kit (Miltenyi Biotec, Bergisch Gladbach, Germany, 130-093-231). Myelin was removed using the MACS Myelin Removal Beads (Miltenyi Biotec 130-096-733). Dissociated cells were resuspended in 500 μL of PBS (azide and serum/protein free) and stained with a fixable cell viability dye (FVD eFluor 780; eBioscience, San Diego, CA, USA, 65-0865-14) according to the manufacturer’s instructions (1 μL FVD/mL cell suspension). Cells were washed with 10% FBS in PBS, then resuspended in 500 μL of MACS buffer (Ca^++^ and Mg^++^ free PBS with 2% FBS and 1 mM EDTA), stained with anti-GLAST-PE (1:50; Miltenyi Biotec 130-098-804), and sorted using the MACS anti-PE MicroBeads UltraPure Kit (Miltenyi Biotec 130-105-639) according to the manufacturer’s instructions. Sorted cells were stained with Hoescht (1 μg/mL, BD Biosciences, San Diego, CA, USA, 33342) for 30 min at room temperature, then resuspended in MACS sorting buffer. 

Cells were analyzed with an Amnis ImageStream Mark II imaging flow cytometer (AMNIS). Cells from the MACS sorted fractions and column flow-through were analyzed. Single-stained controls were collected for the compensation matrix to determine the level of spectral overlap (live nucleated cells were examined for PE-GLAST signal and YFP). The resulting compensated image files were analyzed using IDEAS analysis software (AMNIS, v.6.0). For the analysis, focused single cells were selected based on viability (excluding FVD e780 dye) and the presence of a nucleus (Hoescht-positive). The live nucleated cells were examined for PE-GLAST signal, and the specific and punctate GLAST-PE binding to cells was described using the Bright Detail Intensity feature in IDEAS software v.6.0). The antibody binding patterns were confirmed in composite cell images of bright field and PE fluorescence generated from ImageStream data.

### 2.7. Foot Fault Analysis

Mice were placed onto a metal grid (1 cm spacing) that was suspended 12 inches above a table surface. Animals were allowed to walk around the grid for 3 min and were recorded from below. The number of steps and paw slips made with each forelimb were analyzed, and the difference in foot faults was calculated as follows: %slippage = [(#slips/#steps_ispilesional_) − (#slips/#steps_contralesional_)]. The following inclusion criteria were applied: (i) demonstrated deficit following stroke (defined as ±2 standard deviations from the mean of uninjured controls at PSD 4), (ii) >50 steps taken during the test. These criteria resulted in the exclusion of a total of 5 mice from the study.

### 2.8. Gait Analysis

Gait analysis was performed using the automated Noldus CatWalk XT system (Noldus, Wageningen, The Netherlands). Mice were allowed to traverse a glass walkway illuminated with a light, and a number of gait parameters were measured. Mice were pre-trained to cross the platform for 2–3 consecutive days prior to stroke. A minimum of 3 successful trials was required, wherein the speed during crossing did not vary more than 60%. Footprints were recorded using a high-speed video camera and analyzed using CatWalk XT software (v.10.6). Parameters were normalized to the performance of uninjured mice and compared between both stroke groups and *Cre*-injected shams. Parameters that were significantly different between groups at baseline were excluded from analysis.

### 2.9. Data Acquisition and Statistical Analysis

All experiments were conducted by an experimenter blind to injury and treatment conditions. Data were tested for normality using the D’Angostino and Pearson omnibus normality test or the Kolmogorov–Smirnov test. Unpaired one-tailed *t*-tests were used to compare two groups. One-factor analysis of variance (ANOVAs) followed by post hoc analysis (Tukey’s or Dunnet’s test) was used to compare 3 or more groups. Foot fault outcomes were analyzed using 2-way repeated measures ANOVA, followed by Bonferroni post hoc tests when appropriate. Lesion volume was analyzed using Mann–Whitney U test. Differences were considered significant at *p* < 0.05. Values are presented as mean ± SEM.

## 3. Results 

### 3.1. Transduced Cells Persist Long Term

We designed a strategy to deliver *Neurod1* to GFAP+ cells in the stroke-injured cortex. AAV5::*Neurod1* or AAV5::*Cre* control was injected into the injured cortex of *Cre*-conditional reporter mice (R26R-YFP or R26R-tdTomato) on post-stroke day 7 (PSD7) (Figure 1a,b), a time that reflects the subacute phase of injury. To first determine the profile of transduced cells using this delivery strategy, mice were sacrificed at day 5 post-AAV5::*Cre* injection (PSD12). As predicted, the vast majority of transduced (tdTom+) cells expressed GFAP (Figure 1c,d); a small percentage of cells co-expressed the neuronal marker NeuN. A rare tdTom+ cell co-localized with the microglia/macrophage marker Iba1. None of the tdTom+ cells were neuroblasts (DCX+) or oligodendrocytes (Oligo2+) at 5 days post-AAV delivery.

We predicted that perilesional reactive astrocytes would be transduced following stroke and *Neurod1* treatment. To test this, we isolated GLAST+ astrocytes from the cortex of R26R-YFP mice at 3 days post-AAV (PSD10). The number of transduced (YFP+) cells within this population was analyzed using magnetically activated cell sorting. As shown in Figure 1e, 50% of perilesional GLAST+ astrocytes expressed YFP. We next examined the ectopic expression of *Neurod1* over time in the stroke-injured brain. Using both R26R-YFP and R26R-tdTomato mice, we visualized the colocalization of neural phenotypes in fluorescent positive cells (XFP+) at early (PSD28) and late (PSD63) times post-stroke. We quantified the proportion of transduced cells that co-expressed the NeuN and GFAP markers after *Neurod1* or *Cre* was delivered. On PSD 28, following *Neurod1* injection, we observed 21.2 ± 4.9% of all XFP+ cells expressed NeuN, and 46.9 ± 13.4% expressed GFAP (Figure 1f,g). There were significantly fewer XFP+NeuN+ cells on PSD28 in the AAV5::*Cre*-injected mice (8.3 ± 1.7%, *p* < 0.05) and similar numbers of XFP+GFAP+ cells (59.1 ± 9.1%) (Figure 1f,g). Most interesting, by PSD 63, the relative percentage of NeuN+XFP+ cells in *Neurod1* injected mice was 55.0 ± 2.4%, significantly greater than that of *Cre*-injected brains (40.9 ± 5.7%; *p* = 0.044; Figure 1f,h) and this was significantly increased from PSD28 (*p* = 0.0009). No change was seen in the relative percentage of XFP+GFAP+ astrocytes at PSD28 and PSD63 in any of the groups. Notably, there was a significant decrease in the relative fraction of XFP+GFAP-NeuN- cells at PSD63 in *Neurod1*-injected mice when compared to Cre controls (stroke/Cre_63d, 19.0 ± 2.7 vs. stroke/*Neurod1*_63d, 11.1 ± 1.5; *p* = 0.022). Hence, while the percentage of NeuN+ cells expressing Cre is increasing over time, the relative percentage of XFP+NeuN+ cells is significantly greater with ectopic expression of *Neurod1.*


### 3.2. Functional Recovery Is Observed following Ectopic Expression of Neurod1 

We investigated the impact of ectopic *Neurod1* expression on post-stroke recovery. Stroke-injured mice that received *Neurod1* or *Cre* in the stroke-injured cortex on PSD7 were assessed for motor function using the grid walk task at PSD28 and CatWalk for gait analysis at PSD63. Stroke-injured mice displayed a significant motor deficit on the grid walking task at PSD4 prior to treatment with *Cre* or *Neurod1* (*p* = 0.003, *p* = 0.002, respectively) (Figure 1i). As predicted, uninjured mice and sham controls did not show deficits (Figure 1i). By PSD28, only mice that received *Cre* were recovered to baseline performance [stroke/*Neurod1* = 0.22 ± 0.53 slips (*p* = 0.846); stroke/Cre = 1.58 ± 0.26 slips (*p* = 0.005); sham/Cre, 1.91 ± 0.37 slips (*p* = 0.0004); uninjured controls, −0.15 ± 0.34 slips] (Figure 1i). 

Since foot fault deficits are not sustained at longer survival times [49,50], and because an impairment was detected at PSD28 in sham mice that received cortical injections of *Cre* (Figure 1i), we performed gait analysis at PSD63 to compare long-term outcomes of ectopic *Neurod1* expression [51]. Significant deficits were detected in stroke/*Cre* mice in swing speed (5.5 ± 1.63% change; *p* = 0.026) and time spent on the hindpaw (10.60 ± 3.79% change; *p* = 0.025) (Figure 1j). The gait of *Neurod1*-treated mice was indistinguishable from that of uninjured controls, changing 0.00 ± 2.15% in swing speed (*p* = 0.67) and 2.02 ± 1.79% in time spent on the hindpaw (*p* = 0.87) (Figure 1j). Taken together, these findings reveal that ectopic expression of *Neurod1* is sufficient to improve functional outcomes following stroke as early as 3 weeks post-transduction and maintain recovery to at least 8 weeks post-intervention.

### 3.3. The Percent of Transduced Neurons in the Perilesional Cortex Increases over Time 

We predicted that improved functional outcomes would be correlated with increased numbers of XFP+NeuN+ cells in the perilesional cortex. We quantified the relative percent of transduced perilesional neurons (NeuN+XFP+/NeuN+) at 21 and 56 days post-AAV transduction in stroke-injured and sham-injected mice. In the stroke-injured mice, there were significantly more XFP+ neurons in *Neurod1* (2.78 ± 0.7% and 20.2 ± 3.1%, at day 21 and day 56, respectively) compared to *Cre*-injected brains (0.47 ± 0.26% and 4.67 ± 0.91%, at day 21 and day 56, respectively) (*p* = 0.031 for day 28 and *p* < 0.0001 for day 56) (Figure 2a). Further, the relative percentage of NeuN+XFP+/NeuN+ cells in the stroke-injured mice was significantly greater at PSD63 compared to PSD28 (*p* = 0.030). These findings indicate that the relative percentage of transduced neurons in the perilesional cortex is significantly greater in stroke-injured mice following the ectopic expression of *Neurod1*.

To compare the proportions of transduced neurons in the absence of injury, we delivered *Neurod1* and *Cre* to sham animals and quantified the transduced cells that co-expressed NeuN in the equivalent cortical area and time points examined above. We predicted that sham animals that received *Neurod1* would have more XFP+NeuN+ cells following treatment. As predicted, those mice that received AAV5::*Neurod1* had more XFP+NeuN+ cells compared to controls that did not receive *Neurod1* (*Neurod1*, 8.1 ± 1.5% and 11.8 ± 1.0 % of total neurons at day 21 and day 56, respectively; *Cre*, 1.4 ± 1.0% and 3.9 ± 0.3% at day 21 and day 56, respectively) (*p* < 0.05) (Figure 2a). 

We sought to determine the cellular phenotypes associated with the observed long-term motor recovery. We analyzed the expression of neuronal subtype-specific transcription factors CUX1 (upper-layer neurons, layers 2/3 [52]) and CTIP2 (lower-layer projection neurons, layers 5/6 [53]) at PSD63 in R26R-YFP mice that received *Neurod1* post-stroke (Figure 2b). We observed CUX1 exclusively in YFP+NeuN+ cells in the upper layers. Similarly, CTIP2 was only present in YFP+NeuN+ neurons in the lower layers. We performed immunohistochemistry for c-FOS as an indicator of neuronal activity and found that a subpopulation of the YFP+NeuN+ neurons co-expressed c-FOS within the cortex (Figure 2c). Hence, neurons expressing *Neurod1* are active within the cortex at a time when functional recovery is observed. 

We next asked whether the improved outcomes following *Neurod1* treatment were associated with a change in lesion volume. We compared the brains of stroke-injured *Neurod1* and *Cre*-injected animals at PSD63 and found similar lesion volumes (stroke/*Neurod1* = 0.06 ± 0.03 mm^3^; stroke/*Cre* = 0.07 ± 0.01 mm^3^; *p* = 0.62) (Appendix A). Hence, ectopic expression of *Neurod1* does not lead to a change in lesion volume at long survival times and when functional recovery is observed.

## 4. Discussion

Here, we demonstrate that ectopic expression of *Neurod1* is sufficient for functional improvement following sensory–motor stroke. Previous studies have used AAV delivery to ectopically express TFs in vivo [11,14,15,28,32,40,41,54,55]; however, few have demonstrated improved functional outcomes following injury [41,42]. Using the grid walk task and gait analysis, we demonstrate that ectopic expression of *Neurod1* in the stroke-injured brain leads to functional improvement as early as three weeks post *Neurod1* delivery and that the recovery is sustained long-term.

Studying functional outcomes in rodent stroke models is important for determining the preclinical efficacy of interventions. However, this is difficult due to two main elements: (1) the equivalent cellular processes thought to underlie post-stroke neuroregenerative and neuroplastic processes (thought to recapitulate processes that occur in development to some degree) occur in an abbreviated time frame compared to those observed in the clinic (reviewed by [56]); and (2) spontaneous recovery of most widely validated post-stroke behavioral tests (including those used here) can occur within weeks in rodents models, as opposed to the persistent, attenuated timeframes seen in human stroke survivors. As such, the Stroke Therapy Academic Industry Roundtable Preclinical Recommendations [43] suggest that rodent stroke models include long-term outcomes measuring at least 2–3 weeks after stroke. In this study, our 9-week timeline exceeding this recommendation is considered chronic in terms of rodent stroke models [57], and importantly, it was still able to detect deficits in untreated controls that were significantly improved in *Neurod1*-treated animals. This finding has important implications for preclinical stroke research.

Chen and colleagues [42] examined the effects of ectopic expression of *Neurod1* following stroke and reported similar outcomes while highlighting some important considerations for stroke studies. First, Chen et al. [42] used a dual AAV9 system (where *Neurod1* expression is driven by a CAG promoter) following ET-1 stroke-injured mice and found that 70% of all transduced cells expressed the neuronal marker NeuN at 14 days post-stroke. We observed NeuN expression in 21% of transduced cells over a similar time course using a single AAV5 system, where the expression of *Neurod1* was driven by the GFAP promoter. Further, our studies did not reveal a difference in the lesion volume in our stroke-injured mice that received *Neurod1*, which was significantly reduced following *Neurod1* expression in the previous work. Different strains of mice, the severity and location of the injury, the time course for intervention, concentration, and serotype of the AAV are also likely to play a role in the outcomes. Importantly, we showed that a lower conversion efficiency also results in long-term recovery. A comparison of these studies is critical when developing therapeutics to treat stroke and reflects the complexity of the model and the importance of demonstrating success in more than one pre-clinical model of injury to improve the potential for translational success of novel therapeutics.

Interestingly, we found that the proportion of transduced neurons in *Neurod1*-treated mice was significantly increased compared to *Cre* controls at PSD28, a time at which functional recovery was already observed. Similarly, we show that transduced neurons comprise almost 20% of the perilesional population in *Neurod1*-treated mice and express markers of activity and regional specificity. When interpreting our findings, it is important to note that we also observed reporter-positive neurons in *Cre*-injected mice (i.e., in the absence of *Neurod1* delivery). This is interesting and consistent with previous work showing a degree of non-specific expression from the GFAP promoter [28,34,35,58]. This reveals the importance of future studies aimed at discerning whether improved recovery is mediated by the conversion of astrocytes to neurons or other mechanisms, such as enhanced neuronal cell survival and/or neuroprotection as a result of ectopic *Neurod1* expression. An eloquent study has recently demonstrated that the ablation of induced neurons generated from microglia/macrophages following a stroke that encompasses the striatum (middle cerebral artery occlusion) leads to the loss of functional improvements [59]. As we move forward in the field, a recent position paper that outlines the obligatory controls for demonstrating in vivo neuronal reprogramming will provide important guidance. Bocchi et al. [60] include lineage tracing (neuronal and glial), single-cell transcriptomics studies, and functional assessment of induced neurons as important next steps.

Recent papers are drawing attention to the poorly understood mechanism of action when reprogramming TFs are expressed in the in situ brain [34,60,61]. We demonstrated that the number of transduced neurons increases over time, and this is correlated with improved functional outcomes. This temporal increase could be the result of astrocyte-to-neuron reprogramming, as well as delayed expression of the AAV in neurons due to leakiness of the promoter. These possibilities are consistent with the observation that at early times post-AAV delivery, the vast majority of transduced cells did not express neuronal markers, and over 50% of the cortical astrocytes in the perilesional stroke area were transduced with the AAV. Studies revealing the extent of astrocyte heterogeneity raise the possibility that functional recovery could also be due to a shift in astrocyte cell state due to ectopic expression of TFs [62,63]. Indeed, specifically blocking a subset of reactive astrocytes was shown to be neuroprotective in a model of Parkinson’s disease [64], and ectopic expression of *Neurod1* in astrocytes may specifically decrease this reactive astrocyte population [65].

We chose to use AAV5 to deliver *Neurod1* based on previous work suggesting enhanced astrocyte targeting with this serotype [46]. A number of different viral delivery methods have been used for in vivo reprogramming (e.g., lentivirus, adenovirus, and AAV9 [66]. These have shown varied transduction and reprogramming efficiencies. Different delivery methods may preferentially transduce different astrocyte subtypes or other neural cell types, highlighting the importance of choosing an appropriate strategy. In our study, AAV5 was used to drive *Neurod1* expression under the control of the short GFAP (gfaABC(1)D) promoter. We performed reprogramming at 7 days post-stroke in order to preferentially target cortical astrocytes and not neural stem cells that also express GFAP [67,68,69,70], as this is a time we have previously demonstrated a rapid decline in the number of migratory neural stem cells present in the cortex by this time point [71].

With the goal of realizing the translational potential of gene therapies that deliver neurogenic TFs to treat the injured or diseased brain, a comprehensive understanding of the cellular and molecular basis of functional recovery is an important consideration for future studies exploring gene therapy-based approaches for stroke repair. Herein, we have demonstrated that the AAV5-mediated expression of *Neurod1* driven by the shGFAP promoter leads to post-stroke functional recovery. Our work highlights the promise of this strategy for stroke repair.

## Figures and Tables

**Figure 1 biomedicines-12-00663-f001:**
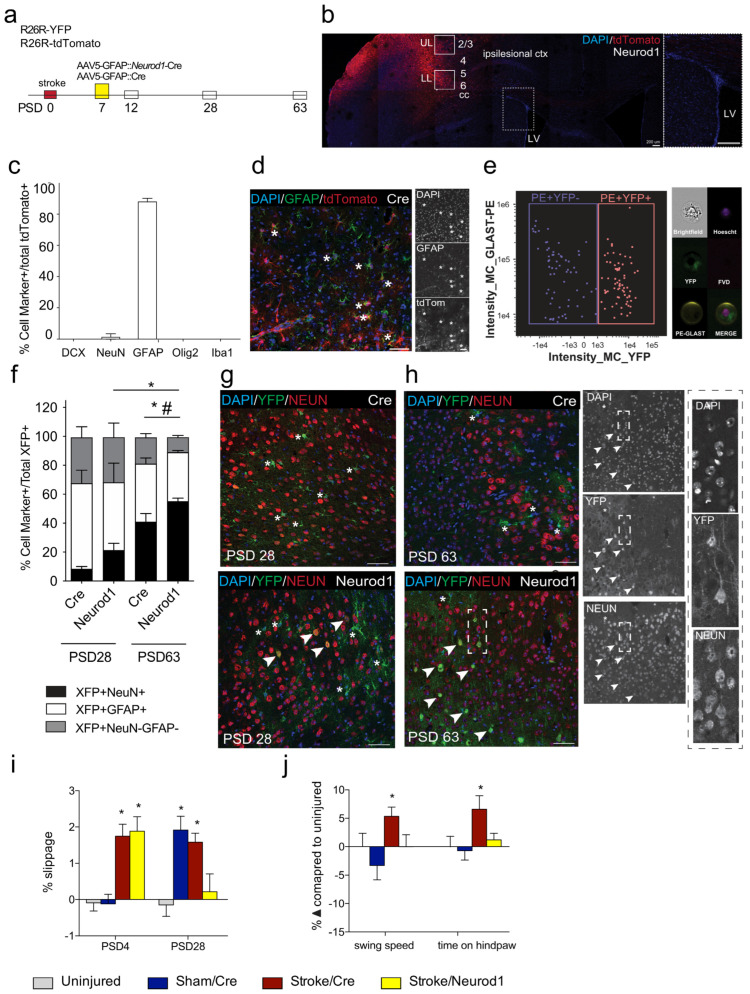
Astrocyte transduction following AAV5-based gene delivery. (**a**) Experimental design and timeline. (**b**) Example image of a stroke-injured cortex 5 days after viral transduction. Dashed box shows enlarged image of the neurogenic subependymal zone (SEZ). (**c**) Quantification of cell type marker expression in transduced cells at 5 days post-*Cre*; *n* = 4/group. (**d**) Example image of transduced astrocytes in a *Cre*-injected brain. Asterisks indicate examples of GFAP+tdTomato+ cells. Scale bar = 100 µm. (**e**) ImageStream plot of MACS sorted GLAST-PE+ astrocytes (40× magnification). Example image of a single (brightfield), live [fixable viability dye (FVD-)], nucleated (Hoechst+), GLAST-PE+YFP+ astrocyte. (1e3 = 1 × 10^3^, 1e4 = 1 × 10^4^, 1e5 = 1 × 10^5^, 1e6 = 1 × 10^6^, −1e3 = 1 × 10^−3^, −1e3 = 1 × 10^−3^) (**f**) Quantification of the percentage of transduced cells that were NeuN+ (black bars), GFAP+ (gray bars), and neither (white bars). * *p* < 0.05, XFP+NeuN+ in *Cre* vs. *Neurod1*; # = *p* < 0.05 XFP+NeuN-GFAP- comparison in *Cre* vs. *Neurod1*; *n* = 3–9/group. Data are expressed as mean ± SEM. (**g**) Example images of *Cre*- and *Neurod1*-treated stroke-injured brains at PSD28 and (**h**) PSD63. Stars = transduced astrocytes; arrowheads = transduced astrocytes expressing NeuN. White dashed box shows enlarged image of transduced astrocytes expressing NeuN. Scale bars = 50 μm. (**i**) Percent slippage in the foot fault test revealed significant impairment in stroke-injured mice prior to reprogramming (PSD4) that was recovered by PSD28 in *Neurod1*-treated mice. A transient deficit was detected at PSD28 in *Cre*-injected sham control mice. *n* = 6–8/group. (**j**) Hindpaw swing speed and time on hindpaw were measured by Catwalk digital gait analysis. At PSD63, *Cre*-injected stroke mice displayed a significant impairment compared to *Cre*-injected sham controls in both parameters. *Neurod1*-treated mice were not significantly different from controls in either task. * *p* < 0.05. *n* = 22–30/group. Data are expressed as mean ± SEM.

**Figure 2 biomedicines-12-00663-f002:**
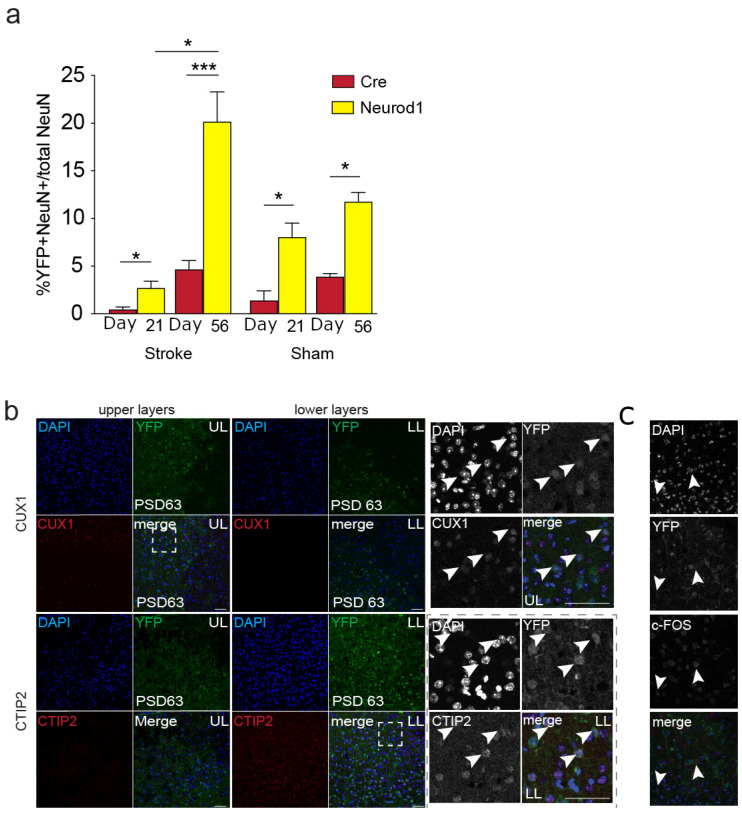
Characterization of cell types produced following AAV5 delivery of *Neurod1*. (**a**) Percentage of XFP+NeuN+ neurons of the total perilesional neurons following stroke or sham injury on day 28 and day 63. * *p* < 0.05, *** *p* < 0.001; *n* = 3–16/group. (**b**) Representative images of CUX1+ and YFP+CTIP2+ neurons in the upper (UL) and lower (LL) layers. White dashed box shows position of enlarged images of CUX1+ cells in the UL and CTIP2+ cells in the LL. Arrowheads indicate examples of layer marker+YFP+ neurons. (**c**) Example image of c-FOS expression in NeuN+YFP+ reprogrammed neurons. Arrowheads indicate examples of c-FOS+YFP+ neurons.

## Data Availability

The data presented in this study are available on request from the corresponding author due to privacy.

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
