# Peer review of "Ectopic Expression of Neurod1 Is Sufficient for Functional Recovery following a Sensory–Motor Cortical Stroke"

_biomedicines, 2024, doi:10.3390/biomedicines12030663_

Round 1

Reviewer 1 Report

Comments and Suggestions for Authors

The authors of the manuscript titled 'Ectopic expression of Neurod1 is sufficient for functional recovery following a sensory-motor cortical stroke' present experimental results to support the finding that ectopic expression of Neurod1 alone in the stroke-injured brain can help improve neuronal repair and in turn associated functional recovery of motor activity. The authors have attempted to address an important treatment paradigm to help stroke-injured brains and validate the approach as an approach to assist patients improve post-stoke impairments. While the motivation and the approach adopted by the authors are sound, there are a few concerns associated with the manuscript which have been listed below.

1. The introduction is confusing as it provides references for previously published research utilizing AAV-mediated ectopic expression of neural TFs but does not clearly explain how the current approach is novel or provides a significant improvement on the previously published results. While the authors claim that they are studying the behavioral and longer-term functional outcomes, it is unclear what the previously published literature has experimented with and observed in terms of functional outcomes as a consequence of ectopic TF expression. The authors seem to oversimplify what is known and what is the gap in knowledge and have missed many scientific publications that have adopted a similar approach with the same motivation and found confounding results. These papers need to be discussed in more detail in the introduction. The authors need to clarify how their strategy helps validate long-term functional outcomes. Especially when defining that the assays were performed 9 weeks post-stroke, how does the timeline translate to long-term when compared to stroke in humans? Provide a relative explanation of experiments in mice compared to humans. Additionally, the authors use the term behavioral outcomes loosely. The motor outcomes measured from the experiments in the paper do not necessarily provide much information on changes in behavior but motor function. Please streamline the verbiage to avoid confusion. 

2. The materials and methods section needs to be improved significantly. It lacks significant details and will prevent the reader from replicating the experiments and performing the analysis listed in the manuscript. For example: the image acquisition parameters are missing, no explanation for the objective selected, if the sample was mounted. What type of colocalization test was performed - Pearsons, Manders? No detail was provided for how the colocalization was quantified. The reader cannot replicate the experiment with the current information provided. 

3. The same holds for the Lesion volume analysis. This section is unclear on how the area was determined. Need to clearly explain the analysis performed using image J and not just state the name of the software. In the section on Astroocyte sorting and imagestream analysis, the authors again fail to clearly explain the parameters employed for image analysis. 

4. All the images provided in the figures seem to be of low resolution. The authors are advised to verify the quality of the images used for generating the figures. The authors are also advised to steer away from the RGB LUTs and adopt the CMYK or non-red-green look-up tables. This will help improve the contrast between the images and help the reader visualize the details the authors want them to focus on. Besides it will also help the readers with red-green color blindness. 

5. Currently, the biggest issue with the presented data is the highly variable number of n for each presented figure subsection. Not all the presented experiments have a decent-sized sample size. The two groups within each graphed experiment tend to have variable sample sizes. Also, the authors are presenting the data as mean +/- SEM. This is a major issue. For example: In figure 1, the data presented has a sample size varying between n=3-9. Given that the results are presented as mean +/- SEM this does not instill much confidence. The authors need to explain the considerably small sample size and the variability when comparing the Cre and Neurod1 sample populations. Figure 1 i again has a small sample size but when it comes to figure 1 j the authors perform the test on a decent-sized sample size (22-30). Can the authors explain the high variability in sample sizes across experiments?

6. For all the images where there is a comparison being depicted, for example figure 1 g,h or figure 2 b, the authors should provide calibration bars for all the channels to verify the images were thresholded equivalently.

7. Lines 227-233: The strategy used to quantify colocalization is not explained. What quantification method was used to verify colocalization? How can anyone interested in replicating the results use the same strategy?

8. Lines 299-308: Explain why CUX1 and CTIP2 were selected as markers for upper and lower layers. In general, when selecting markers explain to the reader the rationale for selecting a protein marker. 

9. Lines 309-314: Grraphs for the quantification not shown. Please provide the appropriate graphs if you wish to make the claim. 

10. Throughout the manuscript, the authors never clearly explain how they qualify the recovery as long-term. The discussion again makes bold claims on the back of a couple of functional recovery experiments with limited sample sizes. The discussion section needs to be toned down to align the one-dimensional approach adopted in the manuscript.

11. The caveats associated with the experimental approach adopted in the manuscript and the limited sample size and supporting evidence to verify functional outcomes need to be better discussed and future directions to validate current findings need to be clearly defined within the discussion section.           

Author Response

Please see the attached the document with the detailed responses to your queries.

We appreciate the feedback.  We believe the manuscript is improved with these revisions.

Reviewer 2 Report

Comments and Suggestions for Authors

It's an interesting study. But I have some suggestions. 

The introduction provides only overly general descriptions of stroke, and it would be beneficial to include an introduction to Sensory-Motor Cortical Stroke.

Additionally, a more detailed introduction to the function of Neurod1 and previous research is needed. It seems unnecessary for the content of Table 1 to be presented as a table; it could be incorporated into the methods section.

The presentation of results, figures, and statistical analyses is too complex and difficult to comprehend. It is recommended to enhance readability.

Further discussion is required on the use of a single AAV5 system for driving Neurod1 expression.

Author Response

Please see the attached document with responses to your queries.

Thank you for your comments.

Round 2

Reviewer 2 Report

Comments and Suggestions for Authors

It has been well revised and I give accept.